# Antimicrobial and Antibiofilm Effect of ε-Polylysine against *Salmonella* Enteritidis, *Listeria monocytogenes*, and *Escherichia coli* in Tryptic Soy Broth and Chicken Juice

**DOI:** 10.3390/foods10092211

**Published:** 2021-09-17

**Authors:** Do-Un Lee, Yeong Jin Park, Hwan Hee Yu, Suk-Chae Jung, Jung-Hee Park, Dae-Hee Lee, Na-Kyoung Lee, Hyun-Dong Paik

**Affiliations:** 1Department of Food Science and Biotechnology of Animal Resource, Konkuk University, Seoul 05029, Korea; ldooo@naver.com (D.-U.L.); pyoungj0509@naver.com (Y.J.P.); yhh0710@naver.com (H.H.Y.); lnk11@konkuk.ac.kr (N.-K.L.); 2Sempio Fermentation Research Center, Sempio Foods Company, Cheongju 28156, Korea; jsukchae@sempio.com (S.-C.J.); pjunghee@sempio.com (J.-H.P.); ldaehee@sempio.com (D.-H.L.)

**Keywords:** ε-polylysine, *Salmonella* Enteritidis, *Listeria monocytogenes*, *Escherichia coli*, antimicrobial effect, antibiofilm

## Abstract

ε-Polylysine (ε-PL) is a safe food additive that is used in the food industry globally. This study evaluated the antimicrobial and antibiofilm activity of antibacterial peptides (ε-PL) against food poisoning pathogens detected in chicken (*Salmonella* Enteritidis, *Listeria monocytogenes*, and *Escherichia coli*). The results showed that minimum inhibitory concentrations (MICs) ranged between 0.031–1.0 mg/mL, although most bacterial groups (75%) showed MICs of 1.0 mg/mL. The reduction in the cell viability of pathogens due to ε-PL depended on the time and concentration, and 1/2 × MIC of ε-PL killed 99.99% of pathogens after 10 h of incubation. To confirm biofilm inhibition and degradation effects, crystal violet assay and confocal laser scanning microscopy (CLSM) were used. The biofilm formation rates of four bacterial groups (*Salmonella*, *Listeria*, *E. coli*, and multi-species bacteria) were 10.36%, 9.10%, 17.44%, and 21.37% at 1/2 × MIC of ε-PL, respectively. Additionally, when observed under a CLSM, ε-PL was found to induce biofilm destruction and bacterial cytotoxicity. These results demonstrated that ε-PL has the potential to be used as an antibiotic and antibiofilm material for chicken meat processing.

## 1. Introduction

Foodborne diseases commonly occur due to food being contaminated by bacteria, bacterial toxins, viruses, parasites, chemicals, and other agents. Researchers have identified more than 250 foodborne diseases [1]. Common symptoms of foodborne diseases include vomiting, nausea, diarrhea, and abdominal cramps. In some cases, severe symptoms can even lead to death. Nearly one in ten people in the world, (approximately 600 million) get food poisoning through contaminated food, and 420,000 people die as a result each year, resulting in 33 million years of disability-adjusted life years. The combined productive and medical costs due to contaminated food in low- and middle-income countries is USD110 billion annually. Food contamination can occur at every stage of the food processing chain, including food production, packaging, and storage [2]. *Salmonella*, *Campylobacter*, enterohemorrhagic *Escherichia coli*, *Listeria*, and *Vibrio cholerae* are among the most common foodborne pathogens [3].

Globally, people enjoy eating poultry meat and its products. However, it is particularly important to pay attention to sanitation as chicken meat and its products can transfer foodborne pathogens [4]. Raw chicken meat is commonly contaminated by *Campylobacter* and occasionally by *Clostridium perfringens* and *Salmonella* [4]. *Listeria monocytogenes* and *E. coli* have also been detected in cooked chicken [5,6]. The global broiler meat production amounted to approximately 100.6 Mt in 2020, and is predicted to grow to approximately 102 Mt in 2021, which is 20 Mt more than what it was in 2012 [7]. In addition, the global consumption of poultry meat is expected to reach 144.87 kt by 2029 [8].

Microorganisms normally live in complicated biological communities, known as biofilms, which provide higher resistance against external physical and chemical attacks such as high temperature, pH, humidity, bactericides, and other antibiotics [9]. Biofilm formation occurs in three steps: bacterial adhesion, maturation, and bacterial dispersion [10]. Adhesion is initiated by the movement of planktonic cells using bacterial nanofibers such as pili and flagella. Subsequently, physicochemical interactions such as electrostatic interaction, hydrophobicity, van der Waals force, and acid-based interactions form bonds between the bacterial cell and the surface [11]. When the attachment is stable, polysaccharides, as the main element of exopolysaccharide (EPS), act as important mediators of attachment. This makes the bacterial adhesion irreversible, creating conditions suitable for maturation [10]. In the maturation stage, bacteria attached to the surface mature and organize a microcolony. Bacteria then perform quorum sensing (QS), an intercellular communication, through various autoinducer secretions [12]. Finally, in the dispersion step, bacteria in the microcolonies disperse when a polysaccharide enzyme disbands the EPS of the biofilm.

Multi-species biofilms are biofilms composed of multiple microbial species. Each microbial species has its own characteristics and reveals some developed and distinct functions on multi-species biofilms that do not exist in their mono-species biofilms [13]. Multi-species biofilms are commonly found in nature, the food industry, and medical devices. They are related to numerous amounts of human bacterial infections, have severe human health impacts, and increase the economic burden on food processing and health-care systems. The intricacy of multi-species biofilms has been associated with their metabolism, interactions, and communication. Additionally, pathogens in multi-species biofilms are more resistant to antimicrobials than those in mono-species biofilms [14].

Antimicrobial peptides (AMPs) are oligopeptides that consist of five or more amino acids [15], and are endowed with a wide range of antibacterial effects against both Gram-positive and Gram-negative bacteria, viruses, and parasites [16]. AMPs occur in all living organisms, ranging from bacteria to plants and animals. AMPs derived from bacteria are known as bacteriocins, and account for 7.36% of the total antibacterial peptide database [16]. Bacteriocins are manufactured by several species to kill other bacterial species for accessing nutrients and space in a limited environment. ε-Polylysine (ε-PL) is a cationic peptide, l-lysine, consisting of 25–35 residues [17]. It is produced by *Streptomyces albulus* and *Lysinopolymerus*, and can inhibit most bacteria, yeast, and viruses as AMPs [18]. In addition, ε-PL has demonstrated a biofilm removal effect [15]. However, research on its antibacterial effects on multiple bacterial pathogens in chicken meat is scarce. In this study, we investigated the antimicrobial and antibiofilm activities of ε-PL against multiple microbial species. In addition, we used chicken juices, the extract of chicken breast, instead of a growth medium, to better understand the relationship between the environment of chickens and the pathogenic bacteria.

## 2. Materials and Methods

### 2.1. Strains, Growth Conditions, and Preparation of Experimental Group

Four biofilm-forming pathogens, *Salmonella* Enteritidis KCCM 12021, *Listeria monocytogenes* H7962 serotype 4, *E**scherichia coli* O157:H4 FRIK 125, and *E. coli* ATCC 25922 were used in this study. The stock cultures of the pathogen species were frozen at −80 °C in broth media supplemented with glycerol (20% *w*/*v*). Stocks were transferred to tryptic soy agar (TSA) twice to activate the strains. *S.* Enteritidis KCCM 12021, *E. coli* O157:H4 FRIK 125, and *E. coli* ATCC 25922 were cultured in tryptic soy broth (TSB; Difco Laboratories, Franklin Lakes, NJ, USA) at 37 °C for 18 h. *L. monocytogenes* H7962 serotype 4 was cultured in TSB supplemented with yeast extract (0.6% *w*/*v*) (TSB-YE) at 37 °C for 18 h.

Four experimental groups were prepared after an 18 h incubation, depending on the bacterial species inoculated. Of these, three were mono-species cultures and the other was a multi-species culture. Each mono-species culture was adjusted to a final concentration of 1 × 10^5^ CFU/mL. The cultures of *E. coli* O157:H4 FRIK 125 and *E. coli* ATCC 25922 were combined in a 1:1 ratio. The multi-species culture was adjusted such that the final concentration of each species was 1 × 10^5^ CFU/mL.

### 2.2. Preparation of Chicken Juice

Raw chicken breast meat was purchased from retail outlets (Seoul, Korea). Chicken juice (CJ) was prepared using the method described by Wang et al. [19] with some modifications. Meats were cleaned of fats and fasciae and then minced using a sterilized knife. Minced chicken meat (100 g) was mixed with 200 mL of sterile water in a filter bag (InterScience, Weymouth, MA, USA). After sealing the mouth of the bag, the filter bag was homogenized using a stomacher (IUL Instruments, Barcelona, Spain) for 90 s. The homogenized mixture was centrifuged at 8000× *g* for 15 min. The supernatant was filtered using a 0.45 μm pore size syringe filter (Advantec, Tokyo, Japan), and the filtered fluid was used as the CJ.

### 2.3. Minimum Inhibitory Concentration (MIC) of ε-PL

The MICs of ε-PL (Shin Seung Hichem Co., Ltd., Seoul, Korea, 50% purity, food grade) were detected using a two-fold dilution method and TSB in 96-well polystyrene microtiter plates (SPL, Gyeonggi-do, Korea) [20]. ε-PL was diluted at double concentrations from 1.95 μg/mL to 8 mg/mL, and 50 μL of each concentration was mixed with 50 μL of bacterial culture adjusted to 1 × 10^5^ CFU/mL in the 96-well plates. The 96-well plates were incubated at 37 °C for 24 h, and the MIC was determined as the lowest concentration of ε-PL that inhibited bacterial growth.

### 2.4. Time-Kill Assay

Time-kill assays were conducted by culturing bacterial cultures in TSB, with the same volume of ε-PL at concentrations of 1/2 × MIC, 1 × MIC, and 2 × MIC at 37 °C, and a non-treated medium was used as a control [21]. Bacterial cultures were prepared in four groups at a final concentration of 1 × 10^5^ CFU/mL each, using the same protocol as described in Section 2.1. Aliquots were collected at 0, 1, 2, 4, 6, 8, and 10 h of incubation and then cultured in appropriate agar media at 37 °C for 24 h, followed by counting viable cell colonies. Single-species bacterial cultures were plated on TSA, and the multi-species bacterial culture was plated on xylose lysine deoxycholate agar (XLD; BD Difco) for *Salmonella*, Oxford agar (MB cell, Seoul, Korea) supplemented with Oxford supplement (MB cell) for *Listeria*, Eosin methylene blue agar (EMB; BD Difco) for *E. coli*, and TSA for total cell counts (*Salmonella + Listeria + E. coli*).

### 2.5. Biofilm Assay

The biofilm assay was conducted using the method described by Yu et al. [22], with some modifications. To investigate the inhibition effect on the formation of biofilms, bacterial cultures were diluted to a final concentration of 1.0 × 10^5^ CFU/mL, and 50 μL of each was mixed in a 96-well plate with volume equivalent to that of ε-PL, with concentrations ranging from 1/2 × MIC to 2 × MIC, using a non-treated TSB as a control. Bacterial cultures were divided into four groups, as described in Section 2.1. Following incubation at 37 °C for 24 h, the bacterial culture was eliminated, and the wells were rinsed twice with sterilized water to remove planktonic cells. Each 96-well plate was dried at 37 °C for 15 min, and 100 μL of 0.1% crystal violet was transferred to each well to stain the biofilms. After 30 min of reaction time, the wells were carefully rinsed with distilled water. Each well was filled with 100 μL of dissolving solution (10% acetic acid and 30% methanol) to dissolve the crystal violet. Optical density (OD) was measured at 570 nm using a microplate reader (Emax, Molecular Devices, Radnor, PA, USA). 

To investigate the degradation effect of ε-PL on mature biofilms, bacterial cultures were adjusted to 1.0 × 10^5^ CFU/mL, and 100 μL of each was incubated in 96-well plates at 37 °C for 24 h. The bacterial suspension was placed in 100 μL of ε-PL at concentrations of 1/2 × MIC to 2 × MIC, followed by incubation at 37 °C for 24 h. The subsequent crystal violet assay was conducted as described above. The biofilm formation rate (%) was determined using the following formula: Biofilm formation rate (%) = OD_treatment_/OD_control_ × 100(1)
where OD_treatment_ and OD_control_ are defined as the absorbance at 570 nm in a dissolving solution.

To investigate the biofilm inhibition and degradation of ε-PL on CJ, the strains were cultured in broth medium at 37 °C for 18 h. The bacterial culture was centrifuged at 12000× *g* at 4 °C for 5 min, and the cell supernatant was discarded. The pellet was washed twice with phosphate-buffered saline (PBS; pH 7.4; Hyclone, Logan, UT, USA) and dissolved in CJ and diluted to a concentration of 1.0 × 10^5^ CFU/mL. ε-PL was diluted in the CJ ranging from 1/2 × MIC to 2 × MIC. The same method was conducted as described above, except that TSB was substituted for CJ.

### 2.6. Hydrophobicity and Auto-Aggregation

The bacterial surface hydrophobicity of food poisoning pathogens was investigated in the presence of ε-PL, following Yang et al. [23] with some modifications. The bacterial culture was adjusted to a concentration of 1.0 × 10^5^ CFU/mL and prepared in four groups, as described in Section 2.1. The bacteria were treated with ε-PL (1/2 × MIC) in TSB at 37 °C for 4 h. Non-treated cells were used as the control.

Cultured cells were centrifuged at 12,000× *g* at 4 °C for 5 min, and the precipitates were rinsed twice. Collected cells were re-dissolved in PBS to adjust the OD_600_ to 0.5 ± 0.05 (OD_initial_). Two milliliters of each suspension was mixed with chloroform (0.5 mL) and vortexed for 2 min. After being static at room temperature for 15 min, the upper aqueous layer was collected and measured at 600 nm (OD_treatment_). The hydrophobicity (%) was calculated using the following equation:Hydrophobicity (%) = (1 − OD_treatment_ / OD_initial_) × 100(2)

Auto-aggregation was conducted using a modification of the method presented by Kim et al. [24]. Cell culture was adjusted to a final concentration of 1.0 × 10^5^ CFU/mL and ε-PL solution at 1/2 × MIC was blended in a ratio of 1:1 and incubated at 37 °C for 24 h. Non-treated cells were used as control. Five milliliters of the mixture was collected and statically incubated at 4 °C for 24 h. After incubation, the upper aqueous layer was measured at 600 nm (OD_treatment_). The sample was then vortexed and measured at 600 nm (OD_initial_). Auto-aggregation (%) was calculated using the following equation:Auto-aggregation (%) = (1 − OD_treatment_/OD_initial_) × 100(3)

### 2.7. Total EPS Production Rate

The total EPS production rate of food-poisoning pathogens was determined according to Song et al. [25], with some modifications. Cell cultures were treated with 1/2 × MIC of ε-PL in TSB at 37 °C for 24 h. Non-treated cells were used as the control. Incubated cells were centrifuged at 8000× *g* at 25 °C for 10 min and the precipitates were dispersed in 1.0 mL of 1.5 M sodium chloride. The cell emulsion was centrifuged at 5000× *g* at 25 °C for 10 min, and 60 μL of the supernatant was mixed with 60 μL of 5% phenol and 4 mL of sulfuric acid by vortexing. Then, the mixtures were incubated at 30 °C for 10 min, and the absorbance was measured at 490 nm. The total EPS production rate (%) was calculated using the following equation:Total EPS production rate (%) = OD_treatment_/OD_control_ × 100(4)
where OD_treatment_ and OD_control_ were defined as the absorbance (OD_490_) of the phenol-sulfuric acid solution, treated with ε-PL and the control, respectively.

### 2.8. Confocal Laser Scanning Microscopy (CLSM) 

CLSM was used to investigate the anti-biofilm effects of ε-PL against biofilm-forming pathogens, as previously described, with some modifications [26]. Before the experiment, to sterilize the glass coupons, glass coupons (2.0 cm × 2.0 cm × 0.2 cm) were sonicated with distilled water 2–3 times and autoclaved at 121 °C for 15 min, followed by air-drying at 60 °C over 2 days. 

Strains were cultured in appropriate broth media at 37 °C for 18 h, and their density was diluted using TSB to adjust the concentration to 1 × 10^5^ CFU/mL for cell suspension. 

To investigate the inhibition effects on biofilm formation, two milliliters of cell suspension was transferred to each well of a 6-well plate (SPL, Gyeonggi-do, Korea) containing glass coupons. Two milliliters of ε-PL solution was adjusted to concentrations of 1 × MIC using TSB and added to each well of the plates. The control experiment used the non-treated medium. The 6-well plates were incubated at 37 °C for 24 h to form bacterial biofilms on the glass coupons. Then, the glass coupons were rinsed twice with PBS to remove planktonic cells. 

To investigate the eradication effects on mature biofilms, four milliliters of cell suspension was added to each well of a 6-well plate with glass coupons. After that, the 6-well plates were incubated at 37 °C for 24 h. Cell cultures were substituted to four milliliters of ε-PL solution at 1 × MIC, and then re-incubated for 37 °C for 24. The subsequent process was the same as that described above.

For CLSM observation, each specimen was dyed as follows: 1 μM/mL each of SYTO9 (Invitrogen™, ThermoFisher Scientific, Carlsbad, CA, USA) and propidium iodide (PI; Invitrogen™, ThermoFisher Scientific) were used to dye the living cells on glass coupons at room temperature for 20 min without light. Following staining, the glass coupons were rinsed twice with PBS and dried at room temperature for 30 min. The specimens were then investigated using a Zeiss LSM 800 microscope (Carl Zeiss, Oberkochen, Germany).

### 2.9. Statistical Analysis 

The results are presented as the mean ± standard error. All experiments were conducted in triplicates. SPSS version 18.0 (SPSS Inc., Chicago, IL, USA) was used for statistical analysis. One-way analysis of variance (ANOVA) was used to determine the significance of the differences.

## 3. Results and Discussion

### 3.1. Antimicrobial Activity

#### 3.1.1. Minimum Inhibitory Concentration

The antimicrobial effect of ε-PL was estimated by investigating the MIC against the chosen foodborne pathogens (Table 1). *S*. Enteritidis KCCM 12021, *L. monocytogenes* H7962 serotype 4, *E. coli* O157:H4 FRIK 125, and *E. coli* ATCC 25922 were selected for this study. These species are representative of food-poisoning pathogens detected in poultry meat. In addition, they can form bacterial biofilms, which are a source of antimicrobial resistance [27,28,29,30]. When *S*. Enteritidis KCCM 12021, *L. monocytogenes* H7962 serotype 4, and *E. coli* were independently cultured, the MICs of ε-PL were 1.0, 0.031, and 1.0 mg/mL, respectively. The MIC of ε-PL in the co-cultured strain was 1.0 mg/mL. ε-PL has been shown to have antimicrobial effects against Gram-positive and Gram-negative bacteria, yeast, and fungi [31,32,33]. Some reports suggest that the optimal MIC levels are 0.1 and 0.05 mg/mL for *S*. Enteritidis and *S*. *typhimurium*, respectively [32]. In addition, *L. monocytogenes* and *E. coli* showed MIC values at 125 and 19.53 μg/mL, respectively [31,33]. These differences can be attributed to the use of the food grade (low purity) and the tested strain. 

#### 3.1.2. Time-Kill Analysis

Figure 1 shows the growth inhibition of foodborne pathogens due to ε-PL. Each strain was cultured independently in TSB and spread on TSA plates. All strains exhibited a concentration-dependent activity. In the control, cell viabilities for *S*. Enteritidis KCCM 12021, *L. monocytogenes* H7962 serotype 4, and the *E. coli* group, cultured at 10 h incubation, reached 8.49 ± 0.09 log CFU/mL, 8.85 ± 0.01 log CFU/mL, and 8.94 ± 0.08 log CFU/mL, respectively. ε-PL at 1 × MIC showed bactericidal activity against all bacteria within 4 h, while at 1/2 × MIC showed bacteriostatic activity against all bacteria within 4 h except to *L. monocytogenes.* The *E. coli* group exhibited a gradual decline on the cell viability depending on incubation time. 

The influence of ε-PL on multi-species cultured bacteria is shown in Figure 2. After co-culturing multiple species in TSB, each species was differentiated by spreading the multi-species culture on a selective agar medium. Multi-species cultures showed a slightly different tendency from mono-species cultures, particularly at 1/2 × MIC. *S*. Enteritidis KCCM 12021 was diminished after 6 h of incubation and was extinguished at 10 h (Figure 2A). The cell viability of *L. monocytogenes* H7962 serotype 4 at 1/2 × MIC reached zero after 2 h (Figure 2B). The cell viability of the *E. coli* group with 1/2 × MIC of ε-PL remained stable over time, and did not significantly increase or decrease (Figure 2C). When all strains were cultured together, 1/2 × MIC of ε-PL killed 99.99% of the pathogens (approximately 1.4 × 10^7^ cells) after 4 h of treatment (Figure 2D). All cell viabilities showed an extinction at 1 × MIC of ε-PL after 2 or 4 h.

ε-PL has a cationic surface in water because of its positively charged amino acid residues, and thus inhibits the growth of pathogens sensitive to it [12]. Generally, cationic ε-PL is able to induce transitions in the structure of the peptidoglycan layer in the cell wall [34] and later interoperate with the cell membrane using electrostatic attraction, which increases the cell permeability [35], subsequently leading to the disruption of the cell membrane. Consequently, water-soluble proteins and ions are released from the cytoplasm. In addition, ε-PL enhances reactive oxygen species production, resulting in the inhibition of metabolism in bacteria [36,37].

### 3.2. Antibiofilm Effect

#### 3.2.1. Crystal Violet Staining Assay

Biofilm inhibition effects have been investigated in the initial stages of biofilm formation [38]. The inhibitory effect of ε-PL on biofilms formed by foodborne pathogens was investigated using the TSB (Figure 3A) and CJ media (Figure 3C). Compared to the control, there was a significant difference in concentrations above 1/2 × MIC, regardless of the culture medium or bacterial species (*p* < 0.05). Figure 3A shows that biofilm formation rates of *S*. Enteritidis KCCM 12021, *L. monocytogenes* H7962 serotype 4, *E. coli*, and multi-species strains at 1/2 × MIC were 28.12%, 35.87%, 60.38%, and 35.28%, respectively. Likewise, in the CJ medium, biofilm formation rates for *S*. Enteritidis KCCM 12021, *L. monocytogenes* H7962 serotype 4, the *E. coli* group, and multi-species strains at 1/2 × MIC were 10.36%, 9.79%, 24.73%, and 21.37%, respectively (Figure 3C). 

Biofilm degradation affects the removal activity of mature biofilms of foodborne pathogens [39]. The experiment was conducted in TSB (Figure 3B) and CJ media (Figure 3D). The biofilm formation rate at 1/2 × MIC in TSB were 37.49%, 91.64%, 17.44%, and 69.83% for *S*. Enteritidis KCCM 12021, *L. monocytogenes* H7962 serotype 4, *E. coli* group, and multi-species strains, respectively (Figure 3B). ε-PL had a significant degradation effect on mature *S*. Enteritidis, the *E. coli* group, and multi-species cell biofilms at concentrations above 1/2 × MIC (*p* < 0.05). However, the *L. monocytogenes* H7962 serotype 4 did not show a significant effect. Biofilm formation levels in the CJ medium were 50.41%, 9.1%, 26.73%, and 24.92% for *S*. Enteritidis KCCM 12021, *L. monocytogenes* H7962 serotype 4, the *E. coli* group, and multi-species strains at 1/2 × MIC (Figure 3D). ε-PL significantly diminished the biofilm formation rate in all groups at concentrations above 1/2 × MIC (*p* < 0.05).

Interestingly, ε-PL had inhibitory and degradative effects against bacterial biofilms at concentrations below the MIC, suggesting that ε-PL is an economical antibiofilm agent. Compared to the inhibitory effects of initial biofilm formation (Figure 3A,C) with the degradation of mature biofilm (Figure 3B,D), the biofilm formation rate in the initial stage was generally lower than that of the mature biofilm, except for the *E. coli* group. In addition, the biofilm formation rate of ε-PL in the CJ medium was generally lower than that in the TSB, for both the inhibitory and degradative effects of the biofilm. As biofilm formation and the bacterial cells remaining were both influenced by carbohydrates [40], there were differences between TSB and CJ.

Some researchers have demonstrated the anti-biofilm effects of ε-PL. ε-PL is a mediator of the inhibition of biofilm formation generated by *Salmonella* sp., which is associated with the potential molecular mechanisms [12]. ε-PL downregulated the expression of genes related to cellulose formation, curli amyloid fiber production, flagella-related motility, and QS, and upregulated genes controlling the synthesis of colonic acid against *Salmonella* sp. Other studies have reported the reduction in biofilms formed by *Salmonella* sp., *P. aeruginosa*, *S. aureus,* and *L. monocytogenes* when treated with ε-PL [41].

#### 3.2.2. Bacterial Surface Properties and EPS Production

Hydrophobicity and auto-aggregation are the principal properties that form biofilms during the initial stage [42]. Cell surface hydrophobicity is known to regulate bacterial adhesion on diverse surfaces [38], while their auto-aggregation contributes to the topography, morphology, and maturation of the zenithal biofilm community. EPS is the main component of biofilms, provides structural stability as well as defense against antibiotics, and promotes antimicrobial resistance [43]. *L. monocytogenes* H7962 serotype 4 treated with 1/2 × MIC of ε-PL showed significantly lower levels of hydrophobicity, auto-aggregation, and EPS production than the control group (without ε-PL), with decreased levels of 6.4%, 21.65%, and 25%, respectively. In addition, the EPS production (%) of multi-species culture at 1/2 × MIC of ε-PL was reduced by 43.71% as compared to that of the control group (*p* < 0.05). These results suggest that ε-PL inhibits the formation of biofilms of *L. monocytogenes* at the initial stage by weakening bacterial surface properties, and this characteristic is identically manifested in a co-culture of *L. monocytogenes* H7962 serotype 4 with Gram-negative strains in a limited area. However, *S. Enteritidis*, and *E. coli* strains did not show significant effects on bacterial surface properties or EPS production (data not shown).

### 3.3. CLSM Observation

The inhibitory and degradation effects of ε-PL on the selective pathogen biofilms were optically analyzed using CLSM (Figure 4). Syto9 and PI differ in terms of the penetration capability of cell membranes [44]. Syto9 could penetrate both dead and live cells because of its high permeability, whereas PI has high molecular mass that it can only penetrate cells with damaged membranes. As shown in the CLSM images, ε-PL caused damage to live biofilm cells of *S.* Enteritidis KCCM 12021, *L. monocytogenes* H7962 serotype 4, and the *E. coli* group. The biofilm of the control showed abundant live colonies (green fluorescence), whereas biofilms treated with ε-PL showed few cells or increasing dead cells (red fluorescence). Some photographs of samples treated with 1 × MIC of ε-PL showed a rare-shaped structure due to the adsorption of ε-PL (Figure 4e,g), whereas few other samples showed death of most cells with no reduction in the total cell count (Figure 4c), which demonstrated the powerful penetrability of ε-PL to deeper biofilms, although the rate of eradication of the biofilm slightly decreased. Nevertheless, these results are worthwhile as green fluorescence (live cells) is rarely observed in both types. Visual observation by CLSM proved that ε-PL could exhibit anti-biofilm activity against *Salmonella*, *Listeria*, and *E. coli*, as also evidenced by the crystal violet assay results of the ε-PL antibiofilm activity (Section 3.2.1).

## 4. Conclusions

This study demonstrated that ε-PL has antibacterial and antibiofilm effects against food poisoning pathogens detected in chicken, such as *S.* Enteritidis, *L. monocytogenes*, and *E. coli.* ε-PL decreased cell viability depending on the time and concentration and induced the collapse of the cell membrane of pathogens, which proved the antibacterial effect of ε-PL. The antibiofilm effect of ε-PL was also evidenced by the reduction in biofilm formation and the destruction of the morphological form, as observed using CLSM. Antibiofilm effects were also observed in CJ. This study will help promote the safety of chicken meat processing, as it imitates the composition and microbiome of chicken meat to obtain more predictable results in contaminated environments. Therefore, ε-PL has the potential to be used as an antibiotic and antibiofilm material in the food industry, while accounting for the sanitary concerns of the consumers.

## Figures and Tables

**Figure 1 foods-10-02211-f001:**
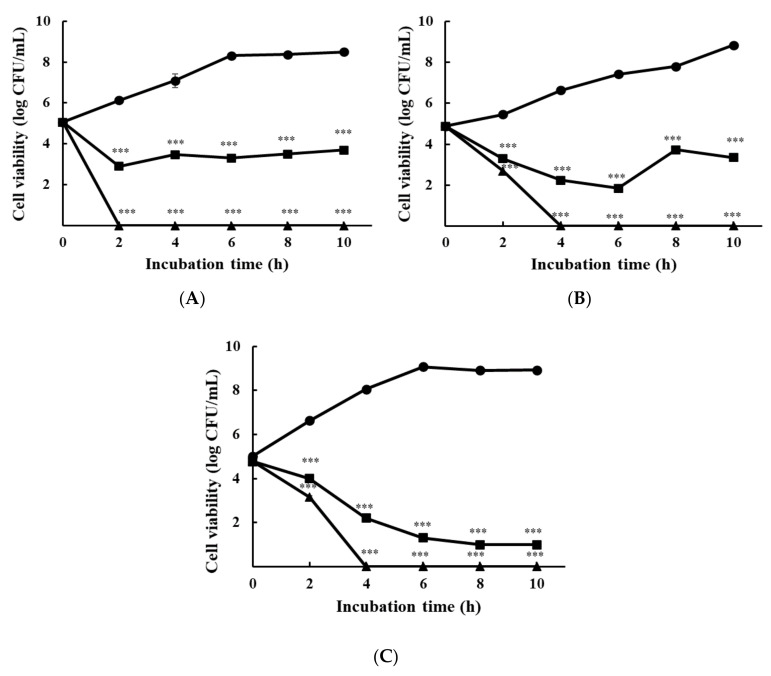
Time-kill analysis of ε-PL at 1/2 × MIC, 1 × MIC, and 2 × MIC against three mono-species cultured bacteria plated on TSA: (**A**) *S*. Enteritidis KCCM 12021; (**B**) *L. monocytogenes* H7962 serotype 4; and (**C**) *E. coli* O157:H4 FRIK 125 + *E. coli* ATCC 25922. All data are expressed as the mean ± standard error. Circle, Control; square, 1/2 × MIC; triangle, 1 × MIC. *** *p* < 0.001, compared to the control group.

**Figure 2 foods-10-02211-f002:**
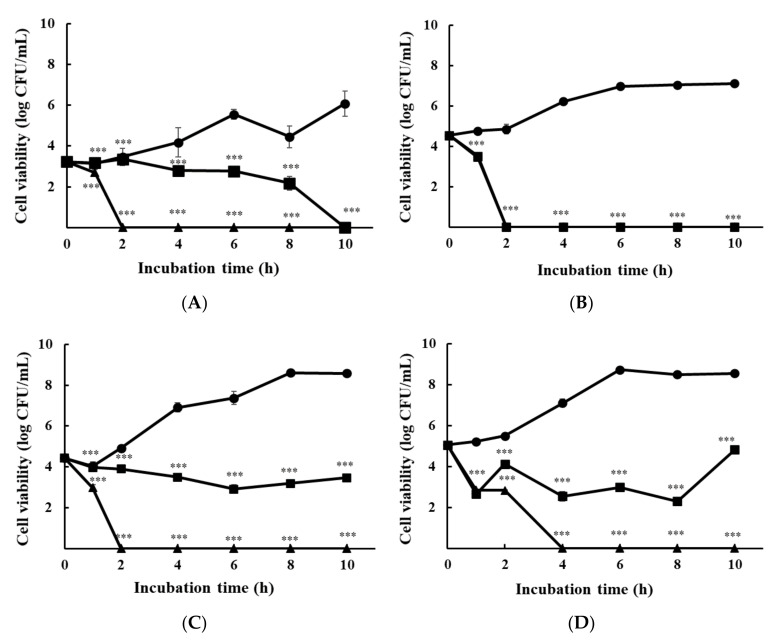
Time-kill analysis of ε-PL at 1/2 × MIC, 1 × MIC, and 2 × MIC against three multi-species cultured bacteria plated on a selective agar medium. Cell viability using (**A**) XLD agar for *S*. Enteritidis KCCM 12021; (**B**) Oxford agar for *L. monocytogenes* H7962 serotype 4; (**C**) EMB agar for *E. coli* O157:H4 FRIK 125 + *E. coli* ATCC 25922; and (**D**) TSA agar for *S*. Enteritidis KCCM 12021 + *L. monocytogenes* H7962 serotype 4 + *E. coli* O157:H4 FRIK 125 + *E. coli* ATCC 25922. All data are expressed as the mean ± standard error. Circle, Control; square, 1/2 × MIC; triangle, 1 × MIC. *** *p* < 0.001, compared to the control group.

**Figure 3 foods-10-02211-f003:**
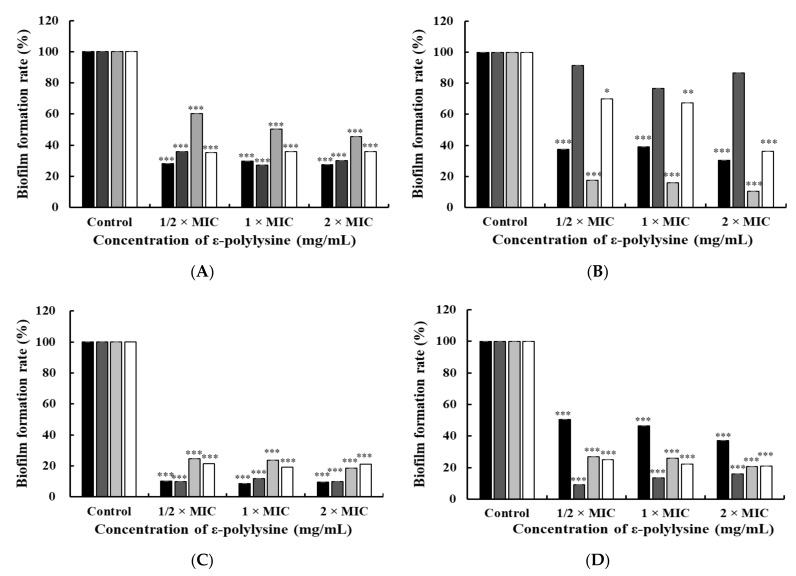
Antibiofilm assay of ε-PL at 1/2 × MIC, 1 × MIC, and 2 × MIC against the bacterial species (*S*. Enteritidis KCCM 12021, *L. monocytogenes* H7962 serotype 4, *E. coli* O157:H4 FRIK 125, and *E. coli* ATCC 25922). (**A**) inhibition of biofilm formation in TSB medium for 24 h; (**B**) degradation of mature biofilm in TSB medium for 24 h; (**C**) inhibition of biofilm formation in the CJ medium for 24 h; and (**D**) degradation of mature biofilm in the CJ medium for 24 h. ε-PL concentrations were relative to the MIC for each organism. All data are expressed as the mean ± standard error and the experiments were conducted in triplicate. Black square, *S*. Enteritidis KCCM 12021; dark gray square, *L. monocytogenes* H7962 serotype 4; light gray square, *E. coli* ATCC 25922 + *E. coli* O157:H4 FRIK 125; white square, *S*. Enteritidis KCCM 12021 + *L. monocytogenes* H7962 serotype 4 + *E. coli* ATCC 25922 + *E. coli* O157:H4 FRIK 125. All data are expressed as the mean ± standard error and the experiments were conducted in triplicate. * *p* < 0.05, ** *p* < 0.01, *** *p* < 0.001, compared to the control group.

**Figure 4 foods-10-02211-f004:**
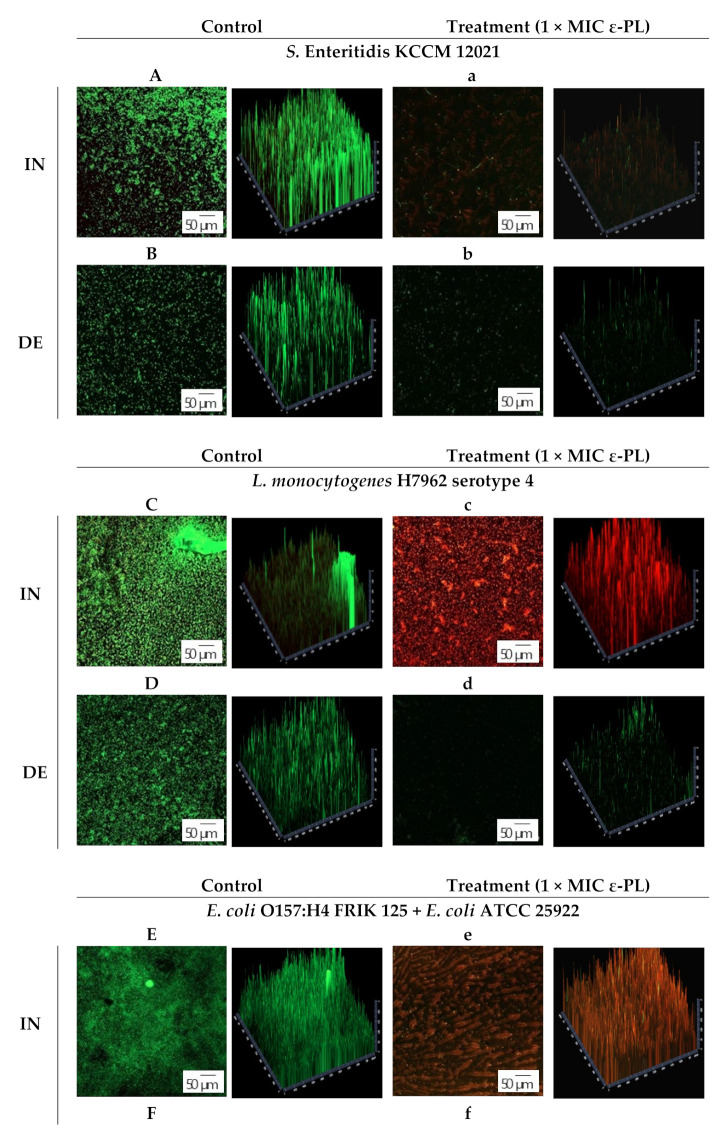
Confocal laser scanning microscopy (CLSM) images of bacterial biofilms cells stained with SYTO 9 (live/green) and PI (dead/red) on glass coupons (100 x magnification). Control and treatment (1 × MIC of ε-PL) are marked in capital letters and small letters, respectively. IN, inhibition of biofilms; DE, degradation of mature biofilms. (**A**,**a**,**B**,**b**) *S.* Enteritidis KCCM 12021; (**C**,**c**,**D**,**d**) *L. monocytogenes* H7962 serotype 4; (**E**,**e**,**F**,**f**) *E. coli* O157:H4 FRIK 125 + *E. coli* ATCC 25922; (**G**,**g**,**H**,**h**); *S*. Enteritidis KCCM 12021 + *L. monocytogenes* H7962 serotype 4 + *E. coli* O157:H4 FRIK 125 + *E. coli* ATCC 25922.

**Table 1 foods-10-02211-t001:** Minimum inhibitory concentration (MIC) values obtained by ε-PL against mono-species cultured strains and multi-species cultured strains.

Serial Number	Strains ^1^				MIC Value (mg/mL)
	*Salmonella* Enteritidis KCCM 12021	*Listeria monocytogenes* H7962 Serotype 4	*Escherichia coli* O157:H4 FRIK 125	*Escherichia coli* ATCC 25922	
1	+	−	−	−	1.0
2	−	+	−	−	0.031
3	−	−	+	+	1.0
4	+	+	+	+	1.0

^1^ +, a strain inoculated; −, a stain not inoculated.

## Data Availability

Not applicable.

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
