# Peer review of "Antimicrobial and Antibiofilm Effect of ε-Polylysine against Salmonella Enteritidis, Listeria monocytogenes, and Escherichia coli in Tryptic Soy Broth and Chicken Juice"

_foods, 2021, doi:10.3390/foods10092211_

Round 1

Reviewer 1 Report

Manuscript "Antimicrobial and Antibiofilm Effect of ε-Polylysine against Salmonella Enteritidis, Listeria monocytogenes, and Escherichia coli in Tryptic Soy Broth and Chicken Juice" presents an interesting antimicrobial substance for use in the meat industry. The manuscript is interesting and very well written. The test methods used are described in great detail. Bacteria are properly selected for this type of research. Correct statistical analyzes are used. The discussion of the results is insightful. The presented research results are very promising and may find practical application. The article is well structured, written in comprehensive language, and appropriately conveys the obtained results as well as the experimental procedure.

Detailed comments:
line 30 - should be Foodborne
Chapter 2.3 - what was the medium used?

Author Response

Manuscript "Antimicrobial and Antibiofilm Effect of ε-Polylysine against Salmonella Enteritidis, Listeria monocytogenes, and Escherichia coli in Tryptic Soy Broth and Chicken Juice" presents an interesting antimicrobial substance for use in the meat industry. The manuscript is interesting and very well written. The test methods used are described in great detail. Bacteria are properly selected for this type of research. Correct statistical analyzes are used. The discussion of the results is insightful. The presented research results are very promising and may find practical application. The article is well structured, written in comprehensive language, and appropriately conveys the obtained results as well as the experimental procedure.

- Thanks for your considering comments. We revised as your comments.

Detailed comments:
line 30 - should be Foodborne

- We revised as your comment (Line 30).

Chapter 2.3 - what was the medium used?

- We used tryptic soy broth (TSB) to dilute ε-PL and cell culture in specific concentration (Line 116).

Reviewer 2 Report

I find the topic of the paper interesting.
In the introductory part, the authors provided enough information to explain the mechanism of food-borne diseases and, in particular, the way in which they survive in different types of food. Thus, they lead us to the goals of this research, ie antimicrobial and antibiotic activity of ε-PL in relation to selected pathogenic microbial species.

The material and methods are written correctly with enough information about the way the experiment was done.

The results are presented in figures, tables, and graphs, and relate to antimicrobial activity, antibiofilm effect, inhibitory results, and ε-PL degradation effect on pathogen biofilms analyzed by CLMS. The results were discussed with relevant research.
In conclusion, the authors state that research has shown antibacterial and antibiofilm effects of ε-PL against food pathogens. As the research was conducted on chicken meat, atuori believe that their results help in promoting food safety or chicken meat production. The potential possibility of using ε-PL should be confirmed in industrial production conditions.

Author Response

I find the topic of the paper interesting.
In the introductory part, the authors provided enough information to explain the mechanism of food-borne diseases and, in particular, the way in which they survive in different types of food. Thus, they lead us to the goals of this research, ie antimicrobial and antibiotic activity of ε-PL in relation to selected pathogenic microbial species.

The material and methods are written correctly with enough information about the way the experiment was done.

The results are presented in figures, tables, and graphs, and relate to antimicrobial activity, antibiofilm effect, inhibitory results, and ε-PL degradation effect on pathogen biofilms analyzed by CLMS. The results were discussed with relevant research.
In conclusion, the authors state that research has shown antibacterial and antibiofilm effects of ε-PL against food pathogens. As the research was conducted on chicken meat, atuori believe that their results help in promoting food safety or chicken meat production. The potential possibility of using ε-PL should be confirmed in industrial production conditions.

- Thanks for your considering comments. We suggested potential possibility in Lab scale (Line 24, 406).

Reviewer 3 Report

The manuscript presents basic information on biofilm inhibiton/disruption using ε-Polylysine, an already established antibiofilm agent.

Major comments:

  1. Additional insights should be provided to strengthen the manuscript.
  2.  Authors have cited the paper https://pubmed.ncbi.nlm.nih.gov/32307570/, but nowhere I found in article discussing in detail about it.  Importantly, the study was done in Salmonella species as well.
  3. Line 335. I don't see authors speaking about Salmonella here also.

MInor comments

1. The figures are low quality and not readable. Better images (graphs) can be provided.

2. Figure alignment is also not constrructive.

3. CLSM images No scale bars.

Author Response

Dear Reviewer,

Thanks for your considering review and comments. We revised as reviewer’s comments and attached the certification by editing service. Please find enclosed, a revised copy of the manuscript ID: foods-1340965R1. In addition, we followed revision checklist for reviewers. We hope the manuscript is now ready for publication.

I request a quick decision because the first author is waiting for the results for her graduation before September 30, 2021.

Thank you for your consideration. I look forward to hearing from you.

Sincerely,

Hyun-Dong Paik

Affiliation and postal address: Department of Food Science and Biotechnology of Animal Resources, Konkuk University, Seoul 05029, Korea

Tel.: +82-2-2049-6011

Email address: hdpaik@konkuk.ac.kr

Reply to reviewer

The manuscript presents basic information on biofilm inhibiton/disruption using ε-Polylysine, an already established antibiofilm agent.

- Thank you for your comments. We revised followed as your requests. As your comments, antibiofilm effects of ε-polylysine established, however, its antibiofilm effects dealt on single strains. We suggested antibiofilm effects using mixed culture and chicken juice. Therefore, this manuscript is significant.

Major comments:

  1. Additional insights should be provided to strengthen the manuscript.

        -  We suggested the importance of this manuscript in Line 82.

  1.  Authors have cited the paper https://pubmed.ncbi.nlm.nih.gov/32307570/, but nowhere I found in article discussing in detail about it.  Importantly, the study was done in Salmonella species as well.

         - We suggested the reference, 12. Shen et al., previously (Line 60, 334).

            Shen, C.; Islam, M.T.; Masuda, Y.; Honjoh, K.I.; Miyamoto, T. Transcriptional changes involved in inhibition of biofilm formation by ε-polylysine in Salmonella Typhimurium. Appl. Microbiol. Biotechnol. 2020, 104, 5427–5436.

  1. Line 335. I don't see authors speaking about Salmonella here also.

         - We revised including Salmonella sp. (Line 335).

Minor comments

  1. The figures are low quality and not readable. Better images (graphs) can be provided.

         - We revised.

  1. Figure alignment is also not constructive.

        - We revised.

  1. CLSM images No scale bars.

        - We added scale bars.

Thanks again for your comments.

Round 2

Reviewer 3 Report

The response seems convincing.